# Incidence of Secondary Sclerosing Cholangitis in Hospitalized Long COVID-19 Patients: A Retrospective Single Center Study

**DOI:** 10.3390/diagnostics14070745

**Published:** 2024-03-30

**Authors:** Christoph R. Werner, Stefano Fusco, Katharina Kienzle, Stefanie Döbele, Kerstin Artzner, Nisar P. Malek, Dörte Wichmann, Siri Göpel

**Affiliations:** 1Department of Gastroenterology, Gastrointestinal Oncology, Hepatology, Infectious Diseases and Geriatrics, University Hospital Tübingen, Otfried-Müller-Strasse 10, 72076 Tübingen, Germany; christoph.werner@med.uni-tuebingen.de (C.R.W.); kerstin.artzner@med.uni-tuebingen.de (K.A.); nisar.malek@med.uni-tuebingen.de (N.P.M.); doerte.wichmann@med.uni-tuebingen.de (D.W.); siri.goepel@med.uni-tuebingen.de (S.G.); 2Institute for Medical Microbiology and Hygiene, University Hospital Tübingen, 72076 Tübingen, Germany; stefanie.doebele@med.uni-tuebingen.de

**Keywords:** secondary sclerosing cholangitis, SSC, SARS-CoV-2, COVID-19, long COVID syndrome, incidence, retrospective observational study

## Abstract

Background: SARS-CoV-2 infection and associated COVID-19 disease can lead to critical illness with a risk of developing a multiple organ failure. Subsequently, this may lead to various pathological sequelae, such as secondary sclerosing cholangitis after surviving COVID-19 (SSC-COVID). Objective: The aim is to retrospectively analyze a cohort of hospitalized patients with first-wave (February 2020–June 2020) SARS-CoV-2 infection and persisting unclear cholangiopathy to determine the incidence of SSC-COVID and its risk factors. Results: A total of 249 patients were hospitalized at the university hospital in Tübingen, Germany, with SARS-CoV-2 infection during the first wave of the pandemic. Of these, 35.3% (88/249) required intensive care treatment; 16.5% (41/249) of them died due to the complications of COVID-19; 30.8% (64/208) of surviving patients could be followed up und were retrospectively analyzed at our center. The incidence of confirmed SSC-COVID was 7.8% (5/64). All SSC-COVID patients had an ICU stay >20 days, for invasive ventilation, positioning treatment, vasopressor treatment, but possible risk factors for SSC were not significant due to the small number of patients. Conclusions: SSC-COVID is an emerging disease in post-COVID patients with a high incidence in our single-center cohort. SSC-COVID should be considered as a differential diagnosis, if unclear cholangiopathy or cholestasis persists after SARS-CoV-2 infection.

## 1. Introduction

From 2019, severe acute respiratory syndrome coronavirus type 2 (SARS-CoV-2) infections and associated Coronavirus Disease 2019 (COVID-19) challenged health care systems worldwide. This SARS-CoV-2 is a “classic” respiratory virus, which primarily infects the upper and lower respiratory system. More recent evidence has shown that coronavirus disease 2019 (COVID-19) is not limited to the airways but can also involve other organ systems, such as skin, kidneys, eyes, and endocrine organs as well as the cardiovascular and central nervous system or the hepatobiliary system [1,2,3,4,5,6,7]. Gastrointestinal symptoms in COVID-19 patients are also common, and liver function impairment including enhanced liver enzymes is frequently observed in acute SARS-CoV2 infection and months later as a variant of long COVID [8]. The severity of the long COVID symptoms, especially of the gastrointestinal disorders varies significantly, depending on comorbidities, physical status, and the age of the patients [9]. There has been a growing focus on cholestatic complications of COVID-19, including sclerosing cholangitis, which are primarily associated with a severe course of COVID-19 [10]. A pronounced biliary tract destruction with the development of secondary sclerosing cholangitis (SSC) in association with COVID-19 pneumonia has been observed, that can be termed “COVID-19 cholangiopathy” [11,12]. The severity of the liver involvement can be seen in the requirement for liver transplantation in certain cases [13].

SSC in general is defined as a structural alteration of the biliary tree, diagnosed with magnetic resonance cholangiopancreatography (MRCP) or endoscopic retrograde cholangiopancreatography (ERCP) and elevated liver enzymes in the absence of other liver diseases, and it may develop in critically ill patients (SSC-CIP) [14]. The incidence of SSC-CIP ranges from 0.5‰ of all ICU patients to under 1% of all hospitalized patients [15,16]. Pathogenesis has not been fully deciphered, but a combination of ischemic injury due to long-term vasopressor treatment and consequential effects leads to microangiopathic reduced perfusion of the small bile ducts; otherwise an increased bile toxicity during critical illness is suspected, which leads to the gradual destruction of the intra- and extrahepatic biliary duct system.

To date, SSC after severe COVID-19 infection (SSC-COVID) has only been described in a few case reports and studies [11,12,16,17,18,19,20]. SSC-COVID is often underdiagnosed because critically ill patients often develop multiorgan dysfunction, which can usually be managed in intensive care (ICU) [15]. The potential differential diagnosis of SSC-COVID is underestimated because certain patients with severe or post-COVID may have intermittent elevations in liver enzymes [11].

Due to the concentration of SSC-COVID patients in our institution, we decided to retrospectively analyze the patients treated during the first wave of COVID-19 to determine the frequency, risk factors, and outcomes of SSC-COVID.

## 2. Materials and Methods

### 2.1. Study Design

The local ethics committee of the University Hospital Tübingen, Germany, approved this study (AZ: 431/2020BO), and it is registered at ClinicalTrials.gov (NCT05233553).

This is a retrospective analysis of hospitalized COVID-19 patients at the University Hospital Tübingen during the first wave of the COVID-19 pandemic in Germany. The study period was February 2020 to June 2020 for patient enrollment in the registry and February 2020 to January 2022 for patient follow-up. 

Inclusion criteria were as follows: (1)Positive SARS-CoV-2 PCR between February 2020 and June 2020;(2)At least one-night stay at university hospital Tübingen;(3)Age ≥ 18 years;(4)At least one follow-up at our outpatient clinic.

During the first wave, our testing policy with respect to SARS-CoV-2 Polymerase chain reaction (PCR) was restrictive due to the limited testing capacities and the absence of antigen tests. At our institution, however, patients with fever and respiratory symptoms or other clinical findings indicative of SARS-CoV-2 such as typical radiological alterations were tested via PCR. 

Exclusion criteria were as follows:(1)Patients of our COVID-19 outpatient clinic, who had an infection with SARS-CoV-2 after June 2020;(2)Treatment of acute COVID-19 infection outside university hospital Tübingen.

Ultimately, 64 of formerly 249 hospitalized patients could thus be included who attended follow-up at our outpatient clinic. See also Figure 1 for a flow-chart of the study design.

Three cases of patients with confirmed SSC after SARS-CoV-2 infection who were not treated at our center nor were infected during another wave were additionally included for case-by-case analysis, without including those patients for statistical analysis on risk factors, neither of SSC-COVID nor elevated liver enzymes, to avoid selection bias.

The focus of the follow-up examinations in our COVID outpatient clinic of patients of the first wave was on cardiopulmonary markers (lung function, cardiac echo). All patients received a routine blood test of whole blood count, kidney function, liver enzymes, inflammation markers. If liver enzymes were elevated (i.e., either Bilirubin, GOT, GPT, AP, GGT, or a combination of those were above the respective upper limit of normal according to our local laboratory), a further evaluation based on the clinical judgement of the physician at the outpatient clinic was performed, depending on the clinical presentation, the severity of the elevation of the liver enzymes, and the patient’s wishes. This included basic serological hepatological diagnostics and further imaging (Ultrasound, MRI, or CT scan). 

Diagnostic criteria for SSC were structural alteration of the biliary tree, diagnosed with magnetic resonance cholangio-pancreatography (MRCP) or endoscopic retrograde cholangio-pancreatography (ERCP), and elevated liver enzymes in the absence of other liver diseases. The establishment of the diagnosis was conducted in synthesis of clinical, radiological, and endoscopic findings [17,18]. In one patient, MRCP was performed only incompletely due to claustrophobia, and the diagnosis was made based on a clinical assessment, as the severity of disease did not require MRCP under anesthesia or more invasive measures such as ERCP or liver biopsy.

The patients matching the inclusion criteria were analyzed with respect to incidence, risk factors, and outcome of SSC-COVID.

Since the number of patients with SSC-COVID was too small, we decided to additionally analyze our cohort with respect to risk factors for elevated liver enzymes, postulating a similar etiology for developing SSC. The patients were analyzed regarding incidence, risk factors, and outcome of SSC-COVID. All hospitalized patients infected with SARS-CoV-2 during the first wave of the pandemic were retrospectively included in this registry. The incidence of SSC-COVID, potential risk factors, and clinical outcome of those patients who were in follow-up (FU) at our COVID outpatient clinic were examined.

### 2.2. Statistical Analysis

Continuous variables were expressed as mean ± standard deviation (SD) or median with interquartile range (IQR). The Mann–Whitney U test was used to examine the differences between the two groups. Categorical variables were expressed as percentages (%) and compared across the groups using Fisher’s exact test. All statistical tests were two-sided, and a *p* value < 0.05 were considered statistically significant. A multivariate analysis by means of logistic regression with stepwise selection (significance level *p* < 0.05) was performed to identify risk factors for the development of SSC-COVID. Only selected variables identified in the univariate analysis were included in the multivariate analysis.

### 2.3. Database

Data are available in SPSS v. 24.0.0.1 (IBM, Armonk, NY, USA) and were presented as mean ± SD.

## 3. Results

During the first wave of SARS-CoV-2 in Tübingen (25 February–30 June 2020), 249 patients were hospitalized with SARS-CoV-2 infection (see Figure 1). Of those, 88 (35.3%) required temporary treatment in the intensive care unit (ICU). Overall, 41 of 249 (16.5%) hospitalized SARS-CoV-2-infected patients died during hospitalization, while 208 (83.5%) were discharged or transferred to secondary hospitals. Of these 208 patients, 144 patients were lost to follow-up, and 64 (30.8%) were seen at our dedicated COVID-19 outpatient clinic at least once for follow-up (see Table 1 for demographic data and details of disease progression). Of those 64 patients, 21 (32.8%) patients had been treated in the intensive care unit, a similar fraction to the whole cohort of hospitalized patients (88/249 in ICU; 35.3%), indicative of a representative cohort with respect to disease severity during hospitalization. 

### 3.1. Incidence of SSC and/or Elevated Liver Enzymes in Hospitalized First Wave COVID-19 Patients

Overall, 5 of these 64 patients were diagnosed with SSC-COVID, representing an incidence of 7.8% in the cohort of patients followed up (see Table 2 for demographic data and disease progression details). For ICU patients in follow-up, the incidence was 23.8% (5/21). Three additional patients with SSC-COVID were not treated at our center during the first wave of SARS-CoV-2 but at other centers or in following waves (for details see Table 2); these patients are discussed separately.

In addition, 15 other patients had elevated liver enzymes without an established diagnosis of SSC-COVID after clinical judgment. Accordingly, the incidence of elevated liver enzymes with or without confirmed SSC-COVID during follow-up in patients after hospitalization for SARS-CoV-2 infection was 32.3% (20/62; in two patients, no liver values were obtained at follow-up).

### 3.2. Characteristics and Outcomes of Patients with SSC-COVID after Hospitalization at Our Center with SARS-CoV-2 Infection

Details of the demographic characteristics of the SSC-COVID patients are shown in Table 2. The mean age of SSC-COVID patients was 70 years (61–82) at diagnosis of infection with SARS-CoV-2. Diagnosis of SSC-COVID was performed using MRCP (3 patients) or a combination of CT scan or MRCP and a therapeutic approach with ERCP (2 patients, see Figure 2). All patients had elevated AP and GGT at follow-up, whereas only one of the five patients had elevated bilirubin at the end of follow-up. All patients had a history of treatment in the intensive care unit during acute infection, including invasive ventilation for more than 20 days (mean 27 days, [21–37 days]), vasopressors, and positioning therapy. One patient required ECMO (21 days). Only one of the five patients who suffered from metabolic syndrome had a diagnosis of steatosis hepatis prior to COVID-19; all other SSC-COVID patients had no known pre-existing liver diseases.

### 3.3. Outcome of Patients with SSC-COVID after Hospitalization at Our Center with SARS-CoV-2 Infection

Two of the patients underwent therapeutic ERCP. In one patient, during ERCP, casts could be extracted and cholestatic markers improved; in the other patient, 3 ERCPs were performed, and although casts could be extracted and a nasobiliary tube was inserted, cholestatic parameters did not improve. Three other patients had a favorable outcome without endoscopic treatment. The patient without improvement in the cholestatic parameters after ERCP died 24 months after the primary infection with SARS-CoV-2. However, since the patient was 84 years old and liver values were stable over time on a high level, probably she died from age. 

### 3.4. Characteristics of Patients with Elevated Liver Values after Hospitalization at Our Center with SARS-CoV-2 Infection

Patients with elevated liver enzymes at follow-up after hospitalization for SARS-CoV-2 infection (20 patients, including 5 with SSC-COVID) showed no significant differences in age, sex, and BMI compared with patients without elevated liver enzymes (see Table 1 for details). Patients with elevated liver enzymes were more likely to have required supplementary oxygen during hospitalization (18/20; 90% vs. 23/42; 54.8%, *p* = 0.007) and had higher rates of intensive care (15/20; 75% vs. 6/42; 14.3%, *p* < 0.001), invasive ventilation (15/20; 75% vs. 5/42; 11.9%, *p* < 0.001), and ECMO (4/20; 20% vs. 1/42; 2.4%, *p* = 0.018) therapy than patients without elevated liver enzymes.

### 3.5. Risk Factors for SSC-COVID or Elevated Liver Values after Hospitalization with SARS-CoV-2 Infection

Due to the small number of SSC-COVID patients, no significant risk factors could be identified for this patient group. Because the number of SSC patients was too small to calculate significant risk factors, we decided to analyze the patients with elevated liver enzymes and compare them with those without elevated liver values for identification of risk factors. For risk factors for the occurrence of elevated liver enzymes during follow-up after hospitalization with SARS-CoV-2 infection, see Table 3. Risk factors included the need for oxygen supply during hospitalization (*p* = 0.013), intensive care treatment (*p* < 0.001), invasive ventilation (*p* < 0.001), ECMO (*p* = 0.044), positioning therapy (<0.001), and treatment with vasopressors (<0.001). Interestingly, duration of invasive ventilation <10 days compared to no invasive ventilation did not have a higher risk of elevated liver enzymes, while duration of invasive ventilation >10 days was a risk factor for developing elevated liver enzymes during follow-up (<0.001; see Table 3 for details). The multivariate logistic regression analysis shows a significance of independent risk factors for ICU stay, for invasive ventilation, for positioning therapy, as well as for vasopressor treatment. 

### 3.6. Characteristics and Outcome of Patients with SSC-COVID Not Treated at Our Center during the First Wave of SARS-CoV-2

Three additional patients with SSC-COVID were treated at our center (see Table 2 for details). One patient had been infected with SARS-CoV-2 during the first wave but treated elsewhere, and the two others were infected during the second and the fourth waves, respectively. They were all referred to our center for liver transplantation. All patients had therapeutic ERCP, either at our institution or elsewhere, without improvement to the cholestatic parameters. One patient was transplanted 8 months after the SARS-CoV-2 infection, one patient died while on waiting list for liver transplantation, and the third patient is still on a waiting list for liver transplantation.

Figure 2 shows the fluoroscopic aspect of a destroyed intrahepatic biliary system during a blocked balloon imaging. There is a sclerosing narrowing of the bigger, central bile ducts and a rarefication of the small, peripheric bile ducts, while the common bile duct is configurated normally. At the bifurcation of the common bile duct, several blurry shapes of the contrast agent can be seen, which correspond to the extracted intraluminal material shown in Figure 3. This is shown in the ERC of patient 1 in Figure 2. 

Figure 3 shows the endoscopic view via duodenoscope during the extraction of solid material from the common bile duct using an extraction balloon. It contains a mixture of sludge and cast. Additionally, a small protective pancreas stent (5 Fr) is seen in the same figure. It is the same patient shown previously in Figure 2. 

Figure 4 shows a nearly vanished intrahepatic biliary system in the fluoroscopy imaging, detecting several contrast agent gaps in the whole biliary system, which corresponds to intraluminal biliary cast material. Even the gallbladder duct appears with strictures. In contrast, the CBD is completely slim the whole length. The roundish unsharp contrast agent “cloud” at the cranial part of the liver, corresponding to liver segment IVa, shows a little biliary abscess. 

Black cast material was removed from the CBD using a biliary extracting basket, as shown in Figure 5. The papilla of Vater is situated at the border of a duodenal diverticulum, which contains further black cast material (seen at the top of Figure 5). 

The MRCP image of Figure 6 points out the multisegmented destruction of the intrahepatic biliary system of the same patient of Figure 4 and Figure 5. Only the CBD presents fully intact with an inconspicuous contrast agent filling. 

The multifocal interrupted contrast agent filling of the intrahepatic biliary tree appears in the MRCP image of patient 5 of Table 2 shown in Figure 7. Figure 7 is a heavily altered intrahepatic biliary duct system of another patient with SSC-COVID. This patient did not have sludge or intraluminal cast material in the bile ducts. 

The ERCP, endoscopic and MRCP findings of patient 3 and patient 4 were similar to the demonstrated figures, so that they are not figured separately. 

## 4. Discussion

SSC associated with SARS-CoV-2 infection (SSC-COVID) is an emerging disease and appears in particular in critically ill patients [11,12,16,19,20,21,22]. We retrospectively analyzed our single-center cohort of hospitalized COVID-19 patients of the first pandemic wave to determine our local incidence of SSC-COVID. In our cohort, the incidence is quite high with a rate of 7.8% (5/64 patients)—referring to all ICU patients in follow-up, the incidence is even higher with 23.8% (5/21). In another study, a high frequency of SSC-COVID was also observed (1/43 invasively ventilated patients [21]), underlining the high risk of developing SSC-COVID in severely affected COVID-19 patients. Secondary sclerosing cholangitis encompasses a group of chronic cholestatic diseases which affect the intra- or extrahepatic bile duct with the risk of evolving into cirrhosis. In the case of SSC-CIP, the development of advanced fibrosis seems to be particularly rapid compared to other etiologies [23]. We also observed a high rate of patients with elevated liver enzymes (32.3%, including the 5 patients with SSC) during follow-up. Since not all formerly hospitalized patients were eligible for follow-up, a potential bias must be discussed with a selection of patients with unfavorable courses being transferred to our outpatient clinic more often than those with complete recovery. Still, referring it to all hospitalized patients, the incidence remains relatively high with 5/88 ICU patients (5.7%) and 5/249 (2%) hospitalized patients.

Although it is rare, the possible causes of SSC-CIP are most probable [15]: a combination of ischemic cholangiopathy due to direct damage to cholangiocytes and indirect damage due to the formation of “toxic bile”. Ischemic cholangiocyte damage is thought to be caused by long-lasting invasive ventilation (high PEEP ventilation, low tidal volumes, and prone positioning), microcirculatory disturbances, and hemodynamic instability. “Toxic bile” formation can be caused by systemic inflammation, which leads to the downregulation of hepatobiliary transporters. Both mechanisms may lead to cholangiocyte necrosis with obliteration of intrahepatic bile ducts and formation of typically biliary casts [24,25]. Additionally, in SSC-CIP drug-induced liver injury may play a role as a third “hit” [26,27]. Accordingly, in the case-by-case analysis of our patients with SSC-COVID, we found that all patients had a prolonged stay in ICU, had invasive ventilation >20 days, received positioning therapy, and received vasopressor treatment. These characteristics also apply to the SSC-COVID patients described in the case reports and series [11,12,16,19,20,21,22]. The average time between the initial infection and the diagnosis of post-COVID-19 cholangiopathy ranges from 90 to 118 days in recent trials [28]. However, the number of SSC-COVID patients in our cohort was too small to calculate significant risk factors. The limitation of our retrospective study is the small number of detected SSC-COVID patients due to a high rate of patients lost due to no follow up. Another limitation is the missing liver biopsies, respectively, and histologies that were taken for SSC diagnosis in other studies, although SSC can also be diagnosed by typical bile duct alterations seen in the MRCP [17,18,29]; whether a biopsy is required for diagnosing SSC might vary from center to center. 

However, the number of SSC-COVID patients in our cohort was too small to calculate significant risk factors. We, therefore, decided to analyze the patients with elevated liver enzymes, assuming the same underlying pathological mechanisms as for SSC-COVID, to compare them with those without elevated liver enzymes for an identification of risk factors for elevated liver enzymes and, thus, possibly for the development of SSC-COVID. In this way, we were able to identify parameters indicative of aggressive, long-lasting critical illness [ICU stay (*p* < 0.001) with vasopressor (*p* < 0.001) and positioning therapy (*p* < 0.001) as well as invasive ventilation ≥10 days (*p* < 0.001)] as risk factors for elevated liver values during follow-up via multivariate logistic regression analysis. Interestingly, a shorter time of invasive ventilation (<10 days) was not associated with elevated liver enzymes in our cohort, suggesting the need for a longer exposure time to the injurious moment. Although elevated liver enzymes may be due to various conditions (e.g., fatty liver, toxic damage, and others), it is probable that the clustering of patients with elevated liver enzymes during follow-up in our post-COVID cohort (31.3% in our cohort, 0.5% in the general population [30]) may indicate an association with COVID-19 itself or the prolonged intensive care treatment. 

These findings show the aggressiveness of SARS-CoV-2 in a certain proportion of infected patients. In our hospital, during the first wave, 35.3% (88/249) of hospitalized patients needed treatment in ICU, and 16.5% (41/249) of hospitalized patients died. 

## 5. Conclusions

Patients with severe COVID-19 including post COVID patients and long COVID patients with persisting elevated liver enzymes and/or new cholestatic liver disease should be considered for a MRCP in order to detect or exclude suspected SSC-COVID. If the bile duct pattern of the MRCP is not typical for the diagnosis of SSC, an ERCP with cholangioscopy and selective biopsies of suspected bile duct endothelial tissue or a transjugular or transcutan liver biopsy should be considered to confirm or exclude SSC-COVID in this patient cohort. Hospitalized patients with SARS-CoV-2 infection and COVID-19 or long COVID with increasing liver enzymes have had a relatively high incidence of SSC in our tertiary center. Possible risk factors of SSC-COVID could be a long-term ICU stay, invasive ventilation, positioning treatment, and vasopressor therapy, but these cannot be confirmed with this study as only five patients exhibited SSC-COVID. More multicentric studies in outpatient care centers are needed to improve the comprehension of this underdiagnosed disease (SSC-COVID). Serum biomarkers are needed to predict or detect SSC-COVID earlier and more often, which should be taken into consideration in future prospective trials, long before severely impaired liver function occurs, requiring a liver transplantation. 

## Figures and Tables

**Figure 1 diagnostics-14-00745-f001:**
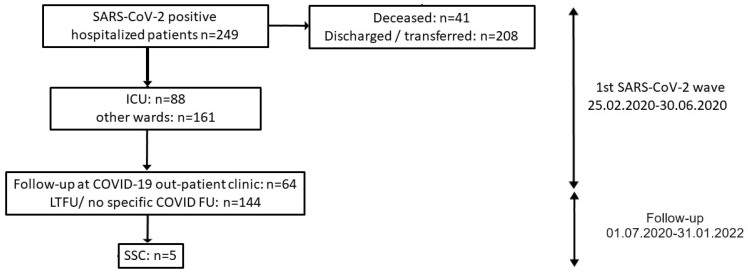
Study design. SSC: secondary sclerosing cholangitis; ICU: intensive care unit; COVID: Coronavirus Disease; LTFU: lost to follow-up; SARS-CoV-2: Severe acute respiratory syndrome coronavirus 2.

**Figure 2 diagnostics-14-00745-f002:**
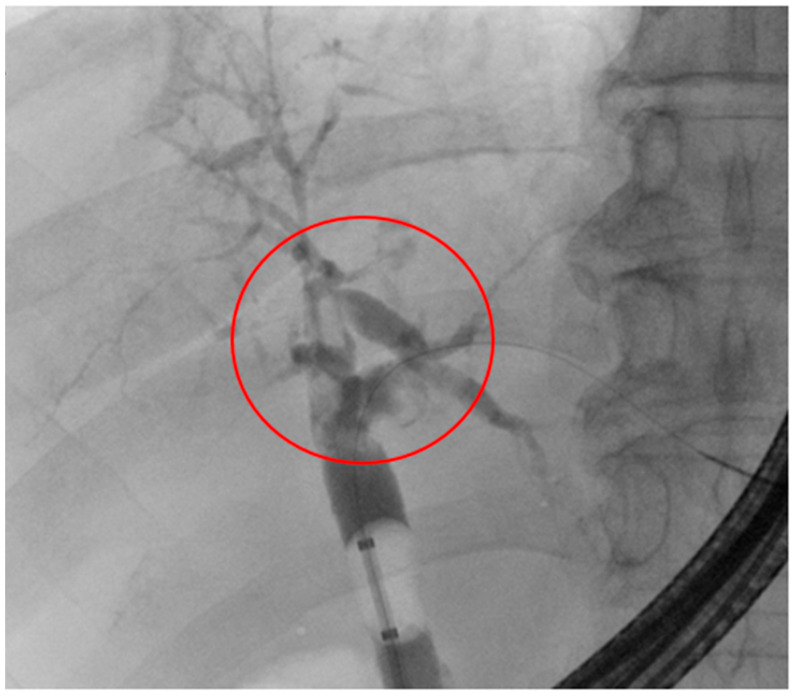
ERC of patient 1 with destroyed intrahepatic biliary system. The most destroyed area is highlighted with the red circle.

**Figure 3 diagnostics-14-00745-f003:**
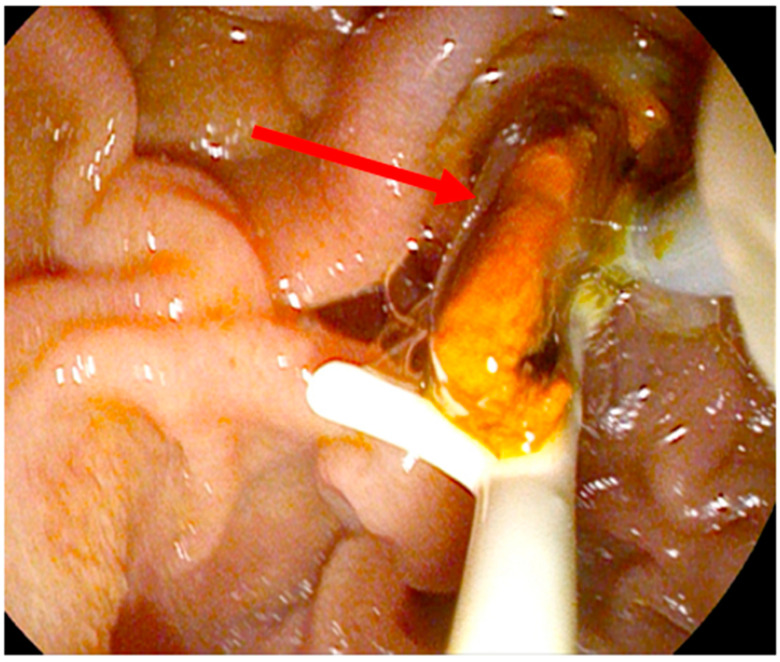
Endoscopic view of patient 1 with extracted sludge and cast (red arrow).

**Figure 4 diagnostics-14-00745-f004:**
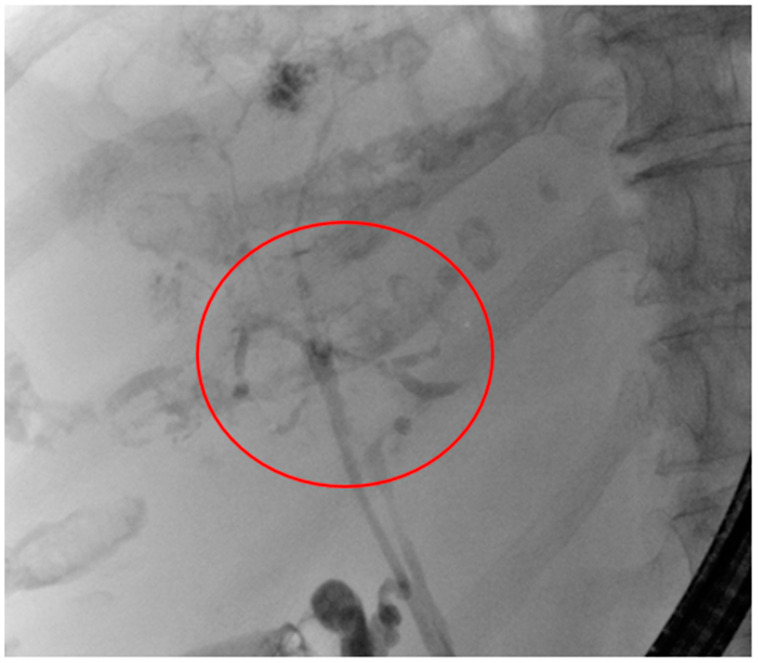
ERC of patient 2 with destroyed intrahepatic biliary system. The most destroyed area is highlighted with the red circle.

**Figure 5 diagnostics-14-00745-f005:**
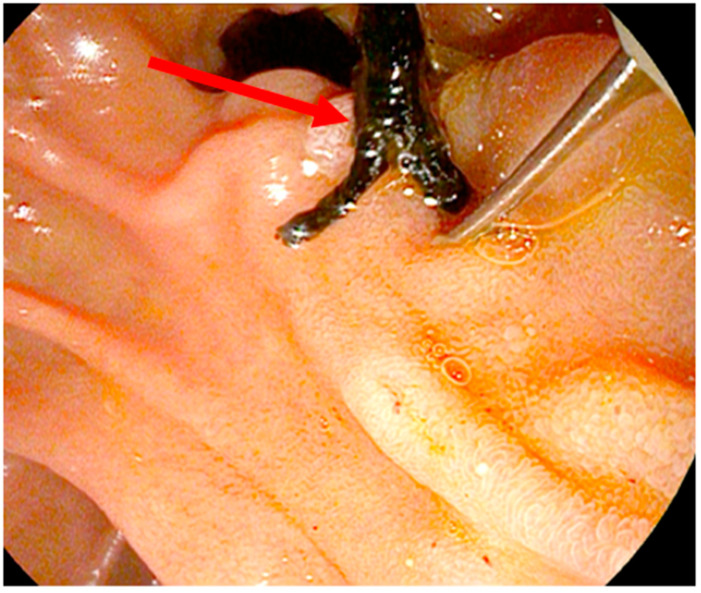
Endoscopic findings in SSC-COVID patient 2. The red arrow shows black cast material.

**Figure 6 diagnostics-14-00745-f006:**
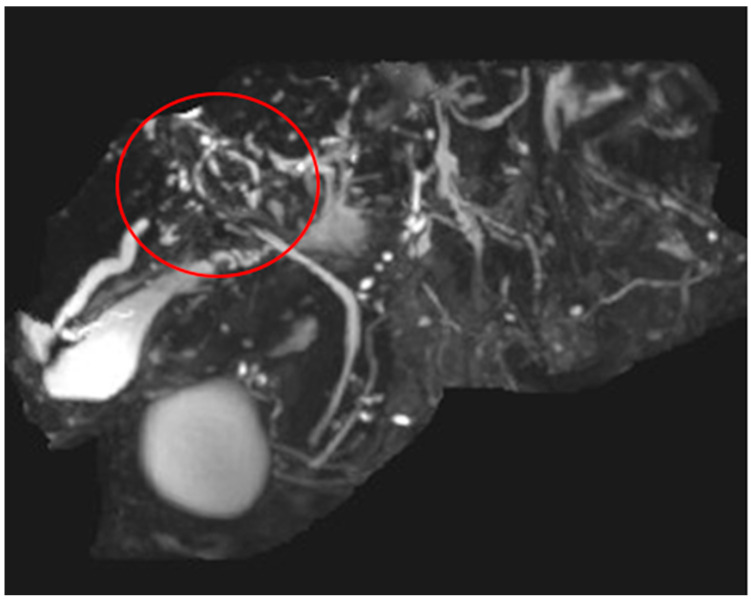
MRCP findings in SSC-COVID patient 2. The most destroyed area is highlighted with the red circle.

**Figure 7 diagnostics-14-00745-f007:**
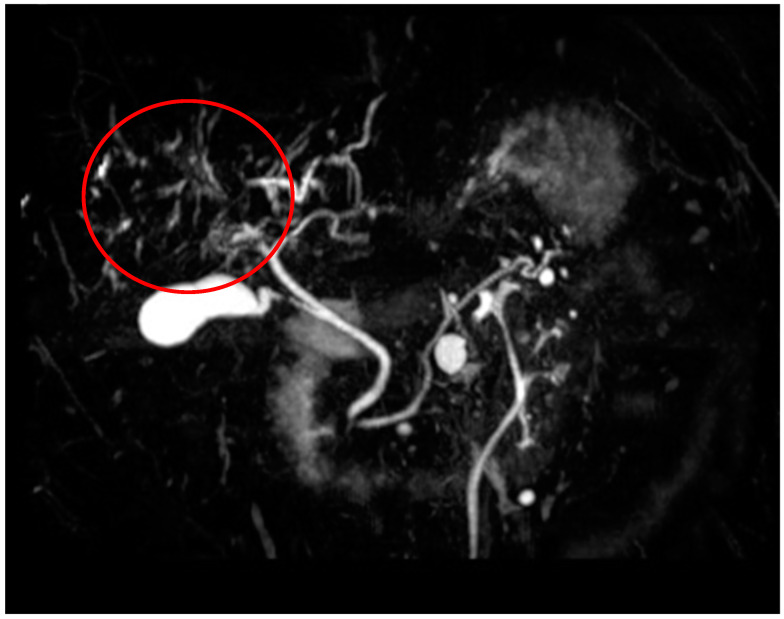
MRCP findings in SSC-COVID patient 5. The most destroyed area is highlighted with the red circle.

**Table 1 diagnostics-14-00745-t001:** Incidence of SSC-COVID, demographics, laboratory values, treatment during course of disease for patients after hospitalization with SARS-CoV-2 infection and for the subset of patients in follow-up with elevated liver values and for those without.

		Total	Elevated Liver Values	No Elevated Liver Values	*p*
Demographics					
Number of patients	n	64	20	42	-
Patients with SSC	n	5 (7.8%)	5 (25%)	0	<0.001
FU (months)	Median IQR Range	4 3–5 2–19	5 3.25–12 3–19	4 3–4.5 2–6	0.001
Age (years)	Median IQR Range	65.5 53.5–77.5 29–87	72.5 55.3–81.8 32–87	62 53–75.3 29–85	0.195
Sex (male)	n (%)	39 (60.9%)	15 (75%)	23 (54.8%)	0.129
BMI (kg/m^2^)	Median IQR Range	27 25–32 18–44	27 26–33 22–40	27 24–32.3 18–44	0s.563
Lab. values at first FU					
Bilirubin (mg/dL)	Median, Range	0.6 (0.2–14.6)	0.6 (0.2–14.6)	0.6 (0.2–1.9)	0.563
GOT/ASAT (IU/L)	Median, Range	17 (6–81)	25 (12–81)	15 (6–44)	<0.001
GPT/ALAT (IU/L)	Median, Range	19.5 (10–113)	35 (17–113)	17.5 (10–40)	<0.001
AP (IU/L)	Median, Range	66 (36–389)	106 (52–389)	60.5 (36–109)	<0.001
GGT (IU/L)	Median, Range	25 (5–686)	147 (16–686)	22 (5–60)	<0.001
Treatment					
Oxygen	n (%)	43 (67.2%)	18 (90%)	23 (54.8%)	0.007
ICU	n (%)	21 (32.8%)	15 (75%)	6 (14.3%)	<0.001
Invasive Ventilation	n (%)	20 (31.3%)	15 (75%)	5 (11.9%)	<0.001
Days Invasive Ventilation	Mean, Range	6.5 (0–47)	16.6 (0–45)	2 (0–47)	<0.001
ECMO	n (%)	5 (7.8%)	4 (20%)	1.1 (2.4%)	0.018
Days ECMO	Mean, Range	1.9 (0–46)	3.75 (0–30)	1.05 (0–46)	0.022
Positioning Therapy	n (%)	20 (31.3%)	15 (75%)	5 (11.9%)	<0.001
Vasopressor Treatment	n (%)	20 (31.3%)	15 (75%)	5 (11.9%)	<0.001

**Table 2 diagnostics-14-00745-t002:** Demographics, laboratory values, treatment during course of disease for all patients with diagnosis of secondary sclerosing cholangitis after hospitalization with SARS-CoV-2 infection.

Patient	1	2	3	4	5	6	7	8
Demographics								
Sex	female	female	male	male	male	male	male	male
Age at baseline	72	82	67	61	68	49	60	49
COVID-19 Wave	1st	1st	1st	1st	1st	1st	2nd	4th
Diagnosis SSC	ERCP, CT	ERCP, MRCP	MRCP	MRCP	MRCP	MRCP, ERCP	MRCP, ERCP	MRCP, ERCP
BMI (kg/m^2^)	31	24	28	37	27	24	19	28
preexisting liver disease	Steatosis	no	no	no	no	no	no	no
Comorbidities	artHTN Adipositas HLP	artHTN	artHTN DM 2	artHTN DM 2 Adipositas	DM 2	Kidney-Pancreas Tx	artHTN	Asthma artHTN DM 2 SAS
FU (months)	9	17	19	13	19	21	15	7
Lab. values								
Bilirubin (mg/dL)	0.6	14.6	1	0.4	0.6	1.5	43.3	11.5
GOT/ASAT (IU/L)	70	57	41	81	18	106	146	85
GPT/ALAT (IU/L)	74	49	38	108	21	138	124	42
AP (IU/L)	208	325	263	130	182	556	451	1499
GGT (IU/L)	206	273	376	619	193	338	276	1110
Treatment								
Oxygen	yes	yes	yes	yes	yes	yes	yes	yes
ICU	yes	yes	yes	yes	yes	yes	yes	yes
Invasive Ventilation	yes	yes	yes	yes	yes	yes	yes	yes
Days of Invasive Ventilation	23	30	37	21	24	9	53	44
ECMO	no	no	no	yes	no	no	no	yes
Days ECMO	0	0	0	21	0	0	0	44
Positioning Therapy	yes	yes	yes	yes	yes	yes	yes	yes
Vasopressor treatment	yes	yes	yes	yes	yes	yes	yes	yes
Outcome	stable	Death	stable	stable	stable	LTx	Death	Waiting for LTx

**Table 3 diagnostics-14-00745-t003:** Predictive model of elevated liver enzymes analyzed for all patients (*n* = 64).

Baseline Parameters	Univariate Analysis		Multivariate Analysis	
	Odds Ratio (95% CI)	Wald *p* Value		
Age				
≥60 years vs. <60 years	1.75 (0.563, 5.44)	0.334		
≥70 years vs. <70 years	1.78 (0.61, 5, 27)	0.285		
Gender				
Male vs. female	2.49 (0.761, 8.07)	0.132		
Oxygen				
Yes vs. no	7.44 (1.53, 36.18)	0.013	1.50 (0.22, 10.08)	0.677
ICU				
Yes vs. no	18 (4.76, 68.1)	<0.001	11.75 (2.64, 52.27)	0.001
Invasive Ventilation (IV)				
Yes vs. no	22 (5.6, 87.99)	<0.001	20.35 (4.65, 89.13)	<0.001
<10 days IV vs. no IV	4.56 (0.39, 53.5)	0.23		
≥10 vs. <10 days or no IV	28.17 (6.15, 128.97)	<0.001	6.05 (0.66, 55.58)	0.112
ECMO				
Yes vs. no	10.25 (1.06, 98.84)	0.044	1.46 (0.12, 17.23)	0.766
Positioning therapy				
Yes vs. no	22.2 (5.6, 87.99)	<0.001	20.35 (4.65, 89.13)	<0.001
Vasopressor				
Yes vs. no	22.2 (5.6, 87.99)	<0.001	20.35 (4.65, 89.13)	<0.001

## Data Availability

Due to ethical restrictions, we cannot share the data publicly.

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
