# Peer review of "Incidence of Secondary Sclerosing Cholangitis in Hospitalized Long COVID-19 Patients: A Retrospective Single Center Study"

_diagnostics, 2024, doi:10.3390/diagnostics14070745_

Round 1

Reviewer 1 Report

Comments and Suggestions for Authors

In this study, Werner et al. reported incidence of secondary sclerosing cholangitis associated with COVID-19 in a single center. SSC-COVID is a relatively new diseased state and not many studies are available. It is important to readers to provide latest information and current situations of SSC-COVID. I do not have any major concerns or criticisms for this manuscript, except some minor suggestions.

·       In the title and the main text, the authors say “high incidence” of SSC-COVID. However, I think this is misleading because the authors reported that the incidence is 7.8%, which I do not feel “high”, and it is unclear that the incidence of SSC-COVID is high compared to what. Please elaborate descriptions to avoid exaggeration, or provide more information to claim clearly that SSC-COVID incidence is high.

·       In discussion, please discuss about limitations and weak points of this study. The authors point out small sample size, but anything else? Diagnosis may be different in different centers so incidence may differ depending on countries or centers? Histological staining to detect liver damage, ductular reaction, or liver fibrosis to confirm conditions and SSC?

·       In Figures 2-7, it would be more understandable for readers if the authors point the area of interest with arrows or circles.

Reviewer 2 Report

Comments and Suggestions for Authors

Werner et al. conducted a retrospective study to determine the incidence of secondary sclerosing cholangitis (SSC) in patients with COVID-19 infection. The main result was a high incidence of SSC related to COVID-19, particularly in ICU-hospitalized patients. The study is interesting; however, several concerns should be clarified to improve the manuscript.

Major Comments:

  1. 1. The introduction section should contain only 3-4 paragraphs. Some paragraphs seem irrelevant and unnecessary. Could the authors consider rewriting them? I suggest they could consist of 1) the burden of COVID-19; 2) gastrointestinal symptoms related to COVID-19; 3) SSC related to COVID-19, including the knowledge gap; and 4) the aim of the study.

  2. 2. The inclusion and exclusion criteria could be rewritten to 1)... 2)

  3. 3. Please add a reference for the diagnostic criteria of this disease.

  4. 4. The last paragraph in the Methods section is repetitive.

  5. 5. Figure 1 could be modified to illustrate the study's flow, including the total number of screening cases, exclusions with reasons, and the final development of SSC related to COVID-19.

  6. 6. Given the study's retrospective nature, calculating the sample size is crucial.

  7. 7. There is not enough detail on statistical analysis. Could the authors provide additional statistical analysis?

  8. 8. The study aimed to determine the differences between patients with SSC from COVID-19 and those without SSC. Therefore, Table 1 should compare columns of SSC vs. non-SSC and describe the variables rather than focusing on elevated vs. non-elevated liver function tests despite the inadequate sample size.

  9. 9. The independent risk factor could be analyzed with multivariable analysis.

Minor Comments:

  1. 1. Abbreviations should be introduced after the full term at their first mention. For example, in line 31 of the Introduction, "COVID-19" should be "Coronavirus Disease 2019 (COVID-19)." Could the authors revise this throughout the manuscript?

  2. 2. There are several incorrect words in the manuscript.
